# Changes in Umbilico–Placental Circulation during Prolonged Intact Cord Resuscitation in a Lamb Model

**DOI:** 10.3390/children8050337

**Published:** 2021-04-26

**Authors:** Kévin Le Duc, Estelle Aubry, Sébastien Mur, Capucine Besengez, Charles Garabedian, Julien De Jonckheere, Laurent Storme, Dyuti Sharma

**Affiliations:** 1Univ. Lille, ULR2694 Metrics—Perinatal Environment and Health, F-59000 Lille, France; capucine.besengez@univ-lille.fr (C.B.); Charles.garabedian@chru-lille.fr (C.G.); julien.dejonckheere@chru-lille.fr (J.D.J.); laurent.storme@chru-lille.fr (L.S.); dyuti.sharma@chru-lille.fr (D.S.); 2CHU Lille, Department of Neonatology, Jeanne de Flandre Hospital, F-59000 Lille, France; sebastien.mur@chru-lille.fr; 3CHU Lille, Department of Pediatric Surgery, Jeanne de Flandre Hospital, F-59000 Lille, France; estelle.aubry@chru-lille.fr; 4CHU Lille, Department of Obstetrics, Jeanne de Flandre Hospital, F-59000 Lille, France; 5INSERM CIC-IT 1403, CHU Lille, 59000 Lille, France

**Keywords:** delayed cord clamping, intact cord resuscitation, umbilico–placental circulation

## Abstract

Some previous studies reported a benefit to cardiopulmonary transition at birth when starting resuscitation maneuvers while the cord was still intact for a short period of time. However, the best timing for umbilical cord clamping in this condition is unknown. The aim of this study was to explore the duration of effective umbilico–placental circulation able to promote cardiorespiratory adaptation at birth during intact cord resuscitation. Umbilico–placental blood flow and vascular resistances were measured in an experimental neonatal lamb model. After a C-section delivery, the lambs were resuscitated ventilated for 1 h while the cord was intact. The maximum and mean umbilico–placental blood flow were respectively 230 ± 75 and 160 ± 12 mL·min^−1^ during the 1 h course of the experiment. However, umbilico–placental blood flow decreased and vascular resistance increased significantly 40 min after birth (*p* < 0.05). These results suggest that significant cardiorespiratory support can be provided by sustained placental circulation for at least 1 h during intact cord resuscitation.

## 1. Introduction

Immediate umbilical cord clamping (ICC) affects cardiopulmonary transition at birth. ICC results in an increase in peripheral systemic vascular resistance and in a decrease in the inferior vena cava flow. Both an increase in afterload and a decrease in preload result in a decrease in systemic blood flow [1].

Conversely, delaying cord clamping between 30 and 180 s after birth increases the transfer of placental blood to the newborn, which increases the circulatory blood volume and prevents the risk of iron deficiency in infants 3 to 6 months old [2,3]. Numerous guidelines recommend delayed cord clamping (DCC) in the newborn infants who did not require immediate resuscitation in order to promote placenta to infant transfusion, although the timing of cord clamping is still debated [4,5,6].

Growing evidence suggests that DCC may promote cardiorespiratory adaptation in newborn infants who require resuscitation at birth. In an experimental study in newborn lambs, we highlighted that placental gas exchange can be sustained after birth, provided special care is taken to prevent compression or kinking of the umbilical cord [7]. Both arterial and venous umbilical blood flow were measured during DCC using Doppler ultrasound in uncomplicated term vaginal deliveries [8]. Venous and arterial umbilical flow were measured for longer than previously described, and this was continued in about half of the babies until the cord was clamped after 5 min [8]. We further showed the feasibility, safety, and effects of intact cord resuscitation in newborn infants with congenital diaphragmatic hernia [9]. Apgar scores were higher and plasma lactate concentrations were lower at 1 h after birth in the infants who were resuscitated, while the umbilical cord remained intact [9]. In this study, the cord was clamped once the baby was intubated and mechanically ventilated at around 7 min. Whether or not umbilico–placental flow can be prolonged any longer is still an open question.

Therefore, we hypothesize that umbilico–placental circulation can be sustained during intact cord resuscitation, provided the cord is secured carefully. To test the hypothesis, we continuously measured the umbilico–placental blood flow and resistance after birth during resuscitation in an experimental model of newborn lambs.

## 2. Materials and Methods

### 2.1. Experimental Model

All animal procedures and protocols used in this study (experimental research protocol n° 2017121218333678) received prior approval from the French Ministry of Agriculture (Ministère de l’Agriculture, de la Pêche et de l’Alimentation) before the study was carried out in the Department of Experimental Research at Lille University (animal experimentation agreement number D59-35010). Pregnant ewes of the Ile de France breed were hosted in individual pens starting a week before and throughout the procedure.

### 2.2. Surgical Procedure

At 138 ± 3 days gestation (term is about 142 days), under sterile conditions and general anesthesia (induction of anesthesia in the ewe with Xylazine–Sédaxylan^®^, 0.05 mg per kg and maintained with 2–5% isoflurane Aerane^®^ in room air/oxygen following intubation) [10,11], the fetal lamb’s left lower limb was exteriorized through a midline laparotomy and hysterotomy of the pregnant ewe.

After fetal analgesia (10 mg IM nalbuphine) and fetal local anesthesia (50 mg SC lidocaine hydrochloride), vascular polyvinyl chloride catheters (4Fr, Vygon^®^ Ecouen, France) were advanced into the aorta after a skin incision in the groin area to measure the aortic pressure. The femoral arterial catheter was inserted to the 8 cm marker, corresponding to the bifurcation of the common umbilical artery of the abdominal aorta. The catheter in the left femoral vein was inserted up to 20 cm into the right atrium.

After a retroperitoneum section and after gentler, blunt dissection of the common umbilical artery at its root from the aorta, a flow transducer (size 6, Transonic System, Ithaca, New York, NY, USA) was placed around the vessel to measure umbilico–placental blood flow (Figure 1).

Then, the lamb was exteriorized from the uterine cavity for exposure of the umbilical cord. In order to measure the umbilical vein pressure, a catheter (20 ga. Arrow) was introduced into one of the two umbilical veins at the base of the cord using the Seldinger technique.

Catheter patency was maintained by a bolus of heparinized saline, 10 UI/mL (Heparine CHOAY^®^ 5000UI, Sanofi-aventis, Paris, France).

At the end of the experimental procedures, animals were euthanized using T61^®^ (Tanax^®^, Intervet Beaucouzé, France: 3 mL/10 kg body weight for the ewe and 0.3 mL/kg for the lamb).

### 2.3. Experimental Design

In order to limit the heat loss and cooling of the lamb, a heat lamp was positioned above the table. The lamb was dried and placed on dry, warm clothes on a table above the ewe’s hooves. Special care was taken to protect the cord from drying and to prevent stretching, kinking, or compressing of the cord.

During the entire resuscitation phase, the sedated pregnant ewe did not receive an injection of oxytocin in order to prevent placental delivery. The ewes are sedated with Isoflurane, and the newborn lambs are being resuscitated while the cord is intact: therefore, the lambs are anesthetized and sedated and do not breathe spontaneously. After pharyngeal suctioning, the lambs were intubated with a 4.5 mm cuffed endotracheal tube (Rüschelit^®^, Teleflex medical, Wayne, PA, USA). First, 20 s of sustained inflation at 35 cmH_2_O was performed, and then the lambs were mechanically ventilated (Infant Star 950^®^ ventilator, Covidien, Dublin, Ireland) in pressure controlled mode (PEEP 5 cmH_2_O, Pmax 24 cmH_2_O, FR 60/min, FiO_2_ 1) for one hour. After shaving the right foreleg and the right hind leg, preductal and postductal SPO_2_ sensors continuously recorded blood oxygen saturation. Mechanical ventilation was adjusted to the target of 40–60 mmHg PCO_2_. If the PCO_2_ was greater than 60 mmHg, inspiratory pressure was increased by 5 cmH_2_O. If PCO_2_ was greater than 80 mmHg, inspiratory pressure was increased by 5 cmH_2_O and respiratory rate was increased by 20 breaths per minute. The SpO_2_ target was between 92 and 99%. FiO_2_ was adapted every 5 min to reach the target. Rectal temperature was continuously recorded during resuscitation.

### 2.4. Hemodynamic and Biological Analysis

The flow transducer was connected to an internally calibrated flowmeter (T201, Transonic Systemes) for continuous measurement of umbilico–placental blood flow (Qup). The zero blood flow value was defined as the flow measured immediately before the beginning of systole. The aortic and the umbilical vein catheters were connected to a pressure transducer monitoring system (Merlin, Hewlett-Packard, Palo Alto, CA, USA). Heart rate (HR) was calculated using the phasic signal from the Qup. Umbilico–placental vascular resistance (Rup) was calculated as the difference between mean aortic pressure (Pao) and mean umbilical vein pressure (Pv) divided by mean umbilico–placental blood flow (Qup).

Blood samples (0.5 mL) from the aortic catheter were taken for blood gas analysis (ABL90 FLEX, Radiometer, Copenhagen, Denmark) 5 min before ventilation, at the beginning of the ventilation, and each 20 min after starting ventilation.

### 2.5. Statistical Analysis

The variables were collected just before and after starting mechanical ventilation. Each lamb served as his or her own control. All statistical analyses were conducted using SPSS Statistic version 24 (IBM corporation, Armonk, NY, USA). Continuous variables were mean ± standard deviation after checking for normal distribution of the data using the Shapiro–Wilk test. The non-parametric Friedman and Wilcoxon distribution-free test were used to assess the significance of differences in respiratory and hemodynamic measures between measurements.

## 3. Results

A total of five fetal lambs (mean weight: 3969 ± 488 g) were used for this experiment. One out of the five lambs was male. Blood gases during the experiment are displayed in Table 1. After 40 min of resuscitation, the blood gases improved with an increase in pH (7.1 to 7.3 at m40, *p* < 0.05) and a normalization of PaCO_2_ (75 to 44 at M40, *p* < 0.05).

Umbilical vein and arterial pressure did not change during the course of the experiment (Table 2).

Throughout resuscitation with an intact cord, a constant placental blood flow of 150 ± 22 mL·min^−1^ at M60 persisted with a statistically significant decrease from M40 compared to M0 (234 ± 75 to 186 ± 79 mL·min^−1^, *p* < 0.05).

Placental vascular resistance gradually increased 1 h after birth, with no statistically significant difference between M0 and M40 (158 ± 10 to 262 ± 111 µW).

## 4. Discussion

Evidence suggests that starting resuscitation maneuvers while the umbilical cord is intact may promote cardiorespiratory adaptation at birth. However, the potential duration of placental circulation after birth is unknown. In the present study, changes in umbilico–placental circulation during intact cord resuscitation were assessed in experiments with fetal lambs. Although umbilico–placental blood flow was sustained until the end of the experiments at 60 min of resuscitation, umbilico–placental blow flow decreased and umbilico–placental vascular resistance increased steadily from 40 min onward. These results suggest that significant cardiorespiratory support can be provided by sustained placental circulation for at least 1 h during intact cord resuscitation.

International resuscitation guidelines currently recommend delaying cord clamping for 60 s in infants who do not need resuscitation, which is largely based on the concept of placenta to infant blood transfusion [12]. Controversies exist regarding the best timing of cord clamping in the newborn infants who need resuscitation. A previous study in newborn infants measured significant blood flow in both the umbilical artery and the umbilical vein for at least 5 min [8]. A previous study by Lefebvre et al. explored the feasibility and effectiveness of intact cord resuscitation in newborn infants with congenital diaphragmatic hernia. In that study, the cord was clamped by 7 min after birth. Apgar scores were significantly higher in the intact cord resuscitation group than in the immediate cord clamp group at 1 min and at 5 min [9]. In a spontaneous breathing lamb model, the cord was clamped at 20 min once the lamb was found to have established a stable breathing pattern [13]. In that study, umbilical blood flow was sustained until cord clamping, although it was transiently reduced during individual breaths.

To the best of our knowledge, our study is the first to explore the potential duration of umbilico–placental circulation after birth during intact cord resuscitation. The blood flow was measured at the root of the umbilical artery before its division into two umbilical arteries. This measure represents the blood flow circulating across the placenta and returning to the lamb via the umbilical vein. The driving force of the placental blood flow is the difference between the aortic pressure and the pressure measured at the distal extremity of the umbilical vein. We found that a maximum of about 250 mL/min of blood crosses the placenta after 10 min of intact cord resuscitation, which may represent 30% of the systemic blood flow of the lamb [14,15,16]. A steady increase of the umbilico–placental vascular resistance beginning 40 min after starting intact cord resuscitation explains the decrease in placental blood flow, as neither aortic nor umbilical vein pressures changed during the 1 h course of the experiment. We have shown that placental blood flow is stable up to 40 min after birth. During pregnancy, the placenta participates in the oxygenation and decarboxylation of the fetus. During delayed cord clamping resuscitation, we showed that PaO_2_ promptly increased and PaCO_2_ decreased. It would be interesting to study the evolution of placental O_2_ uptake (placental VO_2_) during delayed cord resuscitation. Our results indicate that effective cardiorespiratory support can be expected for a prolonged period of time during intact cord resuscitation in the conditions of our experiment.

Oxygen pressure into umbilical arterial blood during pregnancy is around 18 mmHg. In vitro studies on umbilical artery rings indicate that an increase in PO_2_ induces an increase in the vascular tone, with a maximum contraction response observed at a PO_2_ around 280 mmHg [17]. These results suggest that high PaO_2_ during intact cord resuscitation may promote an increase in the umbilical vascular resistance and may contribute to a rapid drop of the umbilico–placental blood flow after birth. In our study, FiO_2_ was adjusted to maintain a physiological SpO_2_. The PaO_2_ was around 60 mmHg during the study period. Our data indicate that an increase in PaO_2_ within the normal range does not impair the umbilical circulation during intact cord resuscitation. Whether or not hyperoxemia may alter the umbilico–placental circulation remains an open question.

In routine clinical practice, oxytocin is infused just after birth for its uterotonic effects in order to prevent postpartum hemorrhage [18]. Oxytocin is known to induce vasoconstriction and increase significantly the uterine and umbilical artery flow resistance during uterine contractions [19]. Furthermore, oxytocin promotes placenta delivery. In our study, oxytocin was not infused at birth to the pregnant ewe after c-section to preserve umbilico–placental resistance. It is likely that oxytocin alters the course of the umbilico–placental circulation during intact cord resuscitation.

During the experiment and the intact cord resuscitation period, the pregnant ewes were anesthetized with isoflurane. Isoflurane has been reported to reduce the contraction of the uterine muscle. Indeed, isoflurane causes uterine relaxation through inhibition of the voltage-dependent calcium channels of the uterine smooth muscle cells [20]. Isoflurane may have delayed normal placental delivery and contributed to sustained placental circulation.

Previous studies showed that gas exchange can be sustained for up to 30 min during Ex utero Intrapartum Treatment (EXIT) procedure [21,22]. EXIT surgery is performed under general anesthesia to achieve uterine relaxation. The fetus is not delivered from the uterus: only the head, neck, and upper part of the thorax is exposed for the maintenance of the uterine volume, thus decreasing the likelihood of uterine contraction and placental abruption. EXIT may be indicated in case of high suspicion of fetal airway obstruction, such as cervical or oral tumors with a consequent high risk of severe fetal hypoxia or death of the newborn infant at birth. EXIT is a complex coordinated procedure that requires a multidisciplinary expert team, including pediatric surgeons, anesthesiologists, obstetricians, and neonatologists.

In the present study, the resuscitation maneuvers are started after birth while the cord is intact. Compared to EXIT, intact cord resuscitation can be used routinely and does not require a multidisciplinary and specialized team. To our knowledge, no study explored the change in umbilico–placental circulation after birth beyond 20 min during intact cord resuscitation.

Our study has limitations. Placenta in sheep (cotyledons) and humans (chorioallantoic) are different. The reproducibility of our physiopathological observations and therefore their extrapolation to the human newborn infants must be done with caution. The study was performed in normal lambs. Further study is required to assess the effects of prolonged intact cord resuscitation in an experimental model of impaired cardiorespiratory adaptation at birth. The present experimental study is a pre-requisite for further studies aiming at assessing the effectiveness of prolonged intact cord resuscitation in cases at high risk for abnormal cardiorespiratory adaptation at birth.

## 5. Conclusions

Our results indicate that effective umbilico–placental circulation can be sustained for up to one hour during intact cord resuscitation in the condition of our experiment including maternal isoflurane anesthesia and excluding oxytocin administration. Our study supports the hypothesis that the duration of intact cord resuscitation may be continued beyond 10 min in order to promote cardiorespiratory adaptation at birth. This information is interesting, because in situations with expected high perinatal mortality such as diaphragmatic hernia and severe hydrops, continuing intact cord resuscitation for up to 1 h could allow better adaptation to extrauterine life.

## Figures and Tables

**Figure 1 children-08-00337-f001:**
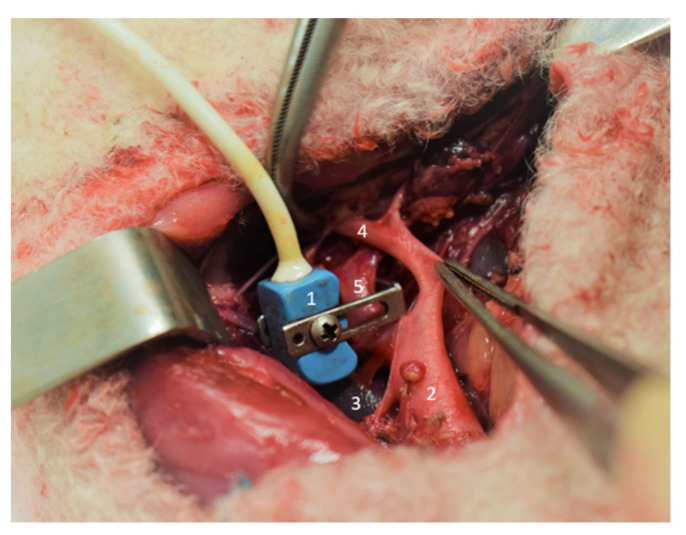
Setting up the probe doppler (**1**): aorta (**2**), vena cava (**3**), iliac artery (**4**), umbilical arteries (**5**).

**Table 1 children-08-00337-t001:** Blood gas during the experiment.

Time	M0	M20	M40	M60
pH	7.1 ± 0.1	7.1 ± 0.2	7.2 ± 0.2	7.3 ± 0.1 *
PaO_2_ (mmHg)	15 ± 4	126 ± 61 *	70 ± 29 *	68 ± 12 *
PaCO_2_ (mmHg)	75 ± 17	62 ± 12	49 ± 11 *	44 ± 8 *
HCO_3_ (mEq.L^−1^)	23 ± 8	20 ± 5	22 ± 6	23 ± 7

Values are means ± SD. PaO_2_, arterial partial pressure of O_2_; PaCO_2_ arterial partial pressure of CO_2_; HCO_3_, bicarbonates. M0 represents the moment when mechanical ventilation was started; M60, the time at which the umbilical cord was clamped. * Significant difference accepted as *p* ≤ 0.05.

**Table 2 children-08-00337-t002:** Respiratory and hemodynamic variables during the experiment.

Time	M-5	M0	M10	M20	M30	M40	M50	M60
HR (Beats/min)	147 ± 40	144 ± 35	133 ± 10	140 ± 20	132 ± 14	132 ± 13	140 ± 14	148 ± 14
PAo (mmHg)	43 ± 4	43 ± 8	48 ± 4	49 ± 4	47 ± 5	51 ± 7	53 ± 8	50 ± 6
Pv (mmHg)	7 ± 3	8 ± 2	8 ± 2	9 ± 2	9 ± 2	8 ± 2	8 ± 2	7 ± 2
Pre-ductal Spo2 (%)	80 ± 20	80 ± 20	98 ± 3	99 ± 2	97 ± 4	97 ± 3	98 ± 3	96 ± 2
Post-ductal Spo2 (%)	38 ± 23	38 ± 23	50 ± 32 *	90 ± 15 *	95 ± 6 *	93 ± 5 *	94 ± 3 *	95 ± 5 *
Qup (mL/min)	232 ± 38	234 ± 75	252 ± 107	214 ± 73	204 ± 98	186 ± 79	148 ± 33 *	158 ± 12 *
Rup (µW)	158 ± 10	151 ± 26	179 ± 66	205 ± 58	224 ± 104	262 ± 111	280 ± 106 *	299 ± 82 *

Values are means ± SD. HR, heart rate; PAo, mean aortic pressure; Pv, umbilical vein pressure; Pre-ductal Spo2, pre-ductal oxygen saturation; Post-ductal Spo2, post-ductal oxygen saturation; Qup, umbilico–placental blood flow; Rup, umbilico–placenta resistance. M0 represents the moment when mechanical ventilation was started; M60, the moment when the umbilical cord was clamped. * Significant difference accepted as *p* ≤ 0.05.

## Data Availability

The data which we presented in the study are available from authors on request. The data are not publicly available due to privacy restrictions.

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
