# Peer review of "Changes in Umbilico–Placental Circulation during Prolonged Intact Cord Resuscitation in a Lamb Model"

_children, 2021, doi:10.3390/children8050337_

Round 1

Reviewer 1 Report

The authors describe briefly the results of experiments on prolonging the umbilical-placental circulation to 60 postnatal minutes, in a newborn lamb model.

The Abstract summarizes the paper accurately.

In the Methods, Figure 1 is unnecessary, not helpful, and it should be removed.

Table 2, currently mislabeled as a second Table 1, contains the same information as figure 4 and figure 3.  It is not clear why figure 4 appears before figure 3.  In any case these figures are superfluous since all the information clearly and efficiently appears in the Table; therefore, these figures should be removed.

The Discussion is a bit verbose, and it tends to repeat some of the findings in the Results more than necessary.  The authors appropriately address the presence of isoflurane and absence of therapeutic oxytocin as potential explanations for the persistence of the umbilical circulation, but they should also address the role of other relevant mechanisms such as oxygen pressure that may be critical in modulating the tone of the umbilical arteries during the normal fetal to neonatal transition.

There are numerous grammatical errors or otherwise awkward expressions in English, which should be corrected in an English version of this paper.  For example:

Lines 17 & 18, should read "experimental neonatal lamb model.  After a C-section delivery, the lambs were resuscitated ventilated for 1 hour while the cord was intact."

Line 29 should read "decrease in preload"

Line 33 is unclear.  To the authors mean risk of iron deficiency?

Line 39, “gas exchange"

Line 47, "while the umbilical cord remained intact"

Line 99, "vena cava"

Line 102, “(20 ga. Arrow)”

Line 105, "catheter patency was maintained"
Line 117 "4.5 mm cuffed endotracheal tube"

Lines 146 and 147, "distribution-free tests were used"

The second table 1 most probably meant to be table 2.  In this stable, 6th row, Oup intended to be Qup, and the same error was made in the legend; for uniformity, the next entry should read “Rup (µW)” (use special character for micron, not a u).

There are many instances throughout the text in which the 2 of O2 or H2 is not subscripted correctly, and in some instances H2O it is written as H20 (zero). Please search and correct all such instances.

In Table 1, the decimal numbers should be expressed with periods, not commas.

Line 205, resistance increased steadily (not increased)
Line 212, "in both the umbilical artery and the umbilical vein"; (delete cord twice)

Line 215, “In that study”

Line 219, “In that study”

Line 260, "study supports the hypothesis"

The formatting of the references is inconsistent and it does not follow the standard of the Children journal.  The issue numbers and dates are in French.  Also, some journal names are abbreviated while others are not, and some article titles are in sentence case (appropriate) while others are in title case.

Reviewer 2 Report

The authors present data on umbilico-placental blood flow and vascular resistance during resuscitation of five healthy term lambs for the duration of 60 minutes. They report that umbilical blood flow can be maintained for up to 60 minutes and that a significant decrease in umbilical blood flow and a corresponding increase in umbilical vascular resistance are not observed until 40 minutes.

It is unclear how the results and conclusions of these experiments could apply to clinical practice. The lambs studied are at term gestation and would otherwise not have required resuscitation had they been delivered naturally.

In the conducted experiments, ewes undergo general anesthesia and lambs are delivered by C-section following instrumentation. Lambs are subsequently intubated and ventilated and hemodynamic parameters including umbilical blood flow, umbilical resistance as well as systemic arterial and venous blood pressure are measured. It is not clear why the lambs needed to be intubated. If utero-placental circulation is maintained, gas exchange should continue to occur at the placental level. This has been clearly demonstrated in humans during EXIT (Ex utero Intrapartum Treatment) and during fetal surgeries (e.g. meningomyelocele repair).

The current experiments do not address whether umbilical-placental circulation would be maintained in a sick subject in need for resuscitation.

Other comments/remarks

Introduction: the authors write “… an increase in afterload and a decrease in reload …” and must have meant “preload”

Introduction: what do the authors mean by "martial deficiency-associated anemia?"

Methods, surgical procedure: catheter permeability, suggest "patency" instead of "permeability"

Methods, experimental design: “injection of oxytocin in order induce placental delivery…” suggest “injection of oxytocin in order to prevent placental delivery…”

Results: The authors report "Placental vascular resistance gradually increased 1 hour after birth, with no statistically significant difference between M0 and M60 (158±10 to 299±82 uW) (Figure 5)." However, in Table 1 and Figure 5, the values at M50 and M60 are denoted with an "*", which in Figure 5 caption is described as a p-value <0.05.

Table 1: what do the "*" represent?

Table 1, row 6: "Oup" should be Qup

Figure 4 shows up before figure 3 in the manuscript text.

Figure 3: can the authors clarify if the values with an asterisk are significant compared to the M0 value?

Figure 4 caption includes "* significant difference accepted as p ≤ 0.05 " However there are no "*" in the figure.

Figure 4: abbreviations PV and PAo should be spelled out. The text and legend state umbilical artery pressure, but in the methodology, arterial pressure was measured in the aorta -authors should clarify this discrepancy.

Discussion, first paragraph: did the authors mean to say umbilico-placental vascular resistance “increased” steadily from 40 minutes... (instead of "decreased")?

Round 2

Reviewer 2 Report

The authors have satisfactorily addressed this reviewer's comments/suggestion in their revised manuscript. 

Author Response

Thank you